# A Usability Study of Classical Mechanics Education Based on Hybrid Modeling: Implications for Sustainability in Learning

Rosanna E. Guadagno [1,*], Virgilio Gonzenbach [2], Haley Puddy [3], Paul Fishwick [4], Midori Kitagawa [5], Mary Urquhart [5], Michael Kesden [5], Ken Suura [5], Baily Hale [5], Cenk Koknar [4], Ngoc Tran [6], Rong Jin [7] and Aniket Raj [8]

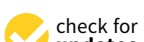

1     Center for International Security and Cooperation, Stanford University, Stanford, CA 94305, USA
2     School of Behavioral and Brain Sciences, University of Texas at Dallas, Richardson, TX 75080, USA; virgilio.gonzenbach@utdallas.edu
3     School of Economic, Political, and Policy Sciences, University of Texas at Dallas, Richardson, TX 75080, USA; haley.puddy@utdallas.edu
4     School of Arts, Technology, and Emerging Communication, University of Texas at Dallas, Richardson, TX 75080, USA; paul.fishwick@utdallas.edu (P.F.); Cenk.Koknar@utdallas.edu (C.K.)
5     School of Natural Sciences and Mathematics, University of Texas at Dallas, Richardson, TX 75080, USA; midori@utdallas.edu (M.K.); urquhart@utdallas.edu (M.U.); kesden@utdallas.edu (M.K.); kds140430@utdallas.edu (K.S.); bgh140130@utdallas.edu (B.H.)
6     Erik Jonsson School of Engineering and Computer Science, University of Texas at Dallas, Richardson, TX 75080, USA; ngoctran@utdallas.edu
7     Computer Science Department, California State University, Fullerton, CA 92831, USA; rong.jin@fullerton.edu
8     Computer Information Technology Department, Lonestar College, Houston, TX 77088, USA; research@aniketraj.com
*     Correspondence: rosannaeg@gmail.com

**Abstract:** A usability study evaluated the ease with which users interacted with an author-designed modeling and simulation program called STEPP (Scaffolded Training Environment for Physics Programming). STEPP is a series of educational modules for introductory algebra-based physics classes that allow students to model the motion of an object using Finite State Machines (FSMs). STEPP was designed to teach students to decompose physical systems into a few key variables such as time, position, and velocity and then encourages them to use these variables to define states (such as running a marathon) and transitions between these states (such as crossing the finish line). We report the results of a usability study on high school physics teachers that was part of a summer training institute. To examine this, 8 high school physics teachers (6 women, 2 men) were taught how to use our simulation software. Data from qualitative and quantitative measures revealed that our tool generally exceeded teacher's expectations across questions assessing: (1) User Experience, (2) STEM-C Relevance, and (3) Classroom Applicability. Implications of this research for STEM education and the use of modeling and simulation to enhance sustainability in learning will be discussed.

**Keywords:** modeling; hybrid modeling; hybrid simulation; usability; sustainability; high school education; physics education; user experience

## 1. Introduction

The United States is a global leader in research and development across the sciences. Unfortunately, this position is threated by an increasing shortage of qualified workers. Indeed, recent data demonstrate this trend—a 2015 study reported that the US ranks low compared to other countries in terms of high school students' science, technology, engineering, and math (STEM) proficiency [1]. To help address this gap, the authors of the present manuscript obtained federal funding to develop a Scaffolded Training Environment for Physics Programming (STEPP) environment for use in high school physics classes. STEPP was designed to be an enjoyable and easy to use modeling and simulation program

that would increase student's physics knowledge, computational thinking, and interest in pursuing a career in STEM. The present paper reports the results of a usability study conducted on a sample of high school physics teachers as a means to understand whether the people likely to use STEPP in their classrooms see STEPP as beneficial to their teaching and likely to meet the aforementioned goals. Please see Supplementary Materials for more details on STEPP.

### 1.1. Physics Teaching Challenges

Students often have misconceptions of physics [2], and these misconceptions are difficult to correct [3]. Although introductory college physics courses are prerequisites for students to enter many STEM fields, college physics professors are often dissatisfied with the preparation that students have received in high school [4]. The performance of U.S. 12th grade physics students was below the international average and among the lowest of the 16 nations that administered this same physics assessment to a comparable population of their students [5]. High school computer science education is in worse condition. Despite the importance of computing in American society, the positive impact this industry has on our economy, and the focus of national, state, and local policy makers on STEM education in the U.S., computer science education in secondary schools has not been growing. Only 47% of all high schools in the United States teach computer science [6]. Among the STEM fields, computer science is the sole subject in the 21st century United States that has actually shown a decrease in the percentage of high school graduates earning credits: down from 25% in 2000 to 19% in 2009 [6]. Furthermore, between 2005 and 2013, the number of secondary schools that offered introductory computer science courses decreased by 4% while the number of secondary schools that offered the Advanced Placement (AP) Computer Science course increased by 6% [7]. In 2011, only 5% of the high schools in the US were certified to teach the AP Computer Science course and only 22,176 students took the AP Computer Science exam nationally [8], while nearly 20 times that number of students took the Calculus AP exams and 6 times that number of students took the Physics AP exams [9]. This concerning trend in computer science education in secondary school is attributed to the certification process and lack of adequate computer science background for computer science teachers across the nation, which is considered confounding [10].

### 1.2. Overview of STEPP

The overarching goal of our NSF-funded STEM + Computing research project is to develop a synergistic scaffolded learning environment in which students learn physics and computational thinking by creating dynamic representations for physics concepts using a 3D video game engine. This paper reports the results of our first usability test of our modules with high school physics teachers.

The modeling methodology used within the implementation of STEPP falls within the domain of hybrid modeling and simulation [11–13]. The hybrid nature of STEPP is in a model that is executed at two levels: the high discrete-event, finite state machine model, and the lower continuous layer beneath each state. States capture the time-based qualitative behavior, and events are Boolean expressions of equational variables for kinematics. Our research plan aligns thematically with previous experimental studies in the simulation community [14]. Rechowicz et al. [15] employ a survey research methodology. The importance of having a human-subject empirical study within model-based systems is emphasized by Giabbanelli et al. [16].

Students often perceive physics as a subject that is too abstract to understand. This is because physics provides knowledge about matter and motion abstracted from our physical world and expressed as mathematical equations [17]. By encoding mathematical equations in a language that computers can process to produce real-time graphical displays, students can turn abstract knowledge back into observable phenomena. Similarly, the game engine provides interactivity and realistic graphics that break down the 3D environment into 1D and 2D environments in a scaffolded approach and incentivizes students to relate physics

concepts to real-world phenomena. Scaffolding will allow students to focus on those aspects of dynamic modeling that are reflexive with physics learning.

Modeling, based on the Finite State Machine (FSM) [18], comes from theoretical computer science as well as state charts used in the Unified Modeling Language (UML) [19]. Both the FSM mathematical structure and the software engineering-based UML serve to emphasize conceptual computer science. Our use of modeling, rather than programming in a language such as Python, is hypothesized to teach the student computational [20] and systems [21] thinking rather than lower-level semantics associated with coding (e.g., text-based programming).

To address these issues, we have developed a Scaffolded Training Environment for Physics Programming (STEPP; see Figure 1), and our research team has partnered with local high school teachers to test three scaffolded tutorial modules that consist of instructions, tutorials, and sample programs that can be incorporated in existing high school physics courses. We maintain that the scaffolded nature of STEPP combined with the method of multiple representations, including color-coded dynamic motions and iterative actions (within modules and between modules) that are in real-time with the simulation, creates a context for sustainable learning.

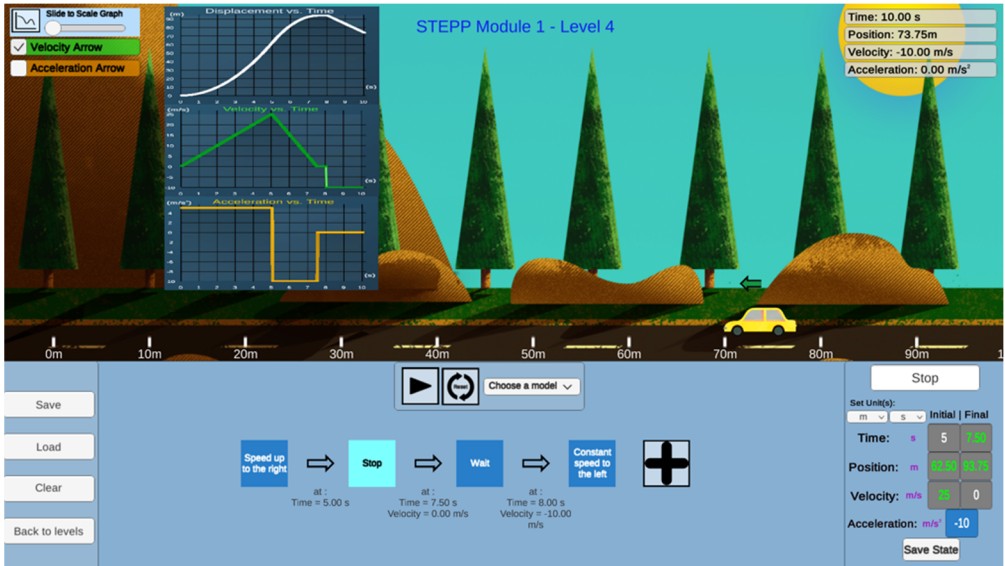

**Figure 1.** Screenshot of STEPP Module 1, Level 4 focused on 1D acceleration as used at the June 2019 summer institute for high school physics teachers.

Our simulation is based on introductory-level classical mechanics in which student users are presented with a graphical simulation containing hybrid discrete-event and continuous models. Users solve specific physics problems that rely on a combination of computational thinking through hybrid modeling, knowledge of Newton's laws of motion, and algebraic problem solving. Our expectation is that when fully deployed, STEPP will increase students' physics knowledge, computational thinking, and interest in a STEM career. Computational thinking is operationally defined as "In the form of a taxonomy of four practices—data practices, modeling and simulation practices, computational problem-solving practices, and systems thinking practices—focusing on the application of computational thinking to mathematics and science" [22]. This important conceptual skill provides people in STEM fields the ability to think like computer scientists [23].

While much of the literature reviewed above pertains to student learning, there is another important group of users that interact with physics learning in US high schools—the teachers. Thus, it was important to assess the usability of STEPP among teachers as well as students. As a result, in June 2019, the STEPP project offered a summer institute at which in-service and pre-service physics teachers would have the first opportunity to learn

and experience STEPP and assist in its further development for use in their classrooms. See Figure 1 for an example screenshot from our modules.

### 1.3. Sustainability in Learning through Modeling and Simulation

What does STEPP and the teaching of introductory physics have to do with sustainability? Sustainability by definition is based on abstract idea of a "system" [24]. System thinking broadens computational thinking to a transferable skill in which automata theory can be applied across disciplines. Other work discusses this need to bridge computational and systems thinking as a path toward sustainability [25].

A system is sustainable if it can maintain its present state and thus avoid a potentially undesirable transition. Although usually associated with ecosystems and the environment, we argue that even elementary mechanics can embody these essential qualities of sustainability. Consider Newton's first law of motion: an object at rest will stay at rest, and an object in motion will stay in motion, unless acted on by an external force.

STEPP is designed to teach the generalizable skills of decomposition and state-based modeling, both of which are critical for building simulations of sustainable systems in a wide variety of contexts. Through a scaffolded instructional environment focused on sustained learning of physics through computer science structures, students are supported in their development of understanding in an accessible environment that contributes to their learning in both disciplines. The sustainable learning goals of STEPP are envisioned to be achieved through the empowerment of users to develop the skills and conceptual understanding necessary for broader self-directed learning and transferability to future applications. The hybrid architecture employed by STEPP in which systems are described by both higher- and lower-level states is described in greater detail in Section 2.3 below. By using STEPP to model the motion of objects with FSMs, we expect that students will build skills in decomposition and state-based modeling that can later be transferred to the modeling and simulation of a variety of sustainable systems.

Overall, the evidence suggests that the current American educational system has not been successful in providing sufficient opportunities for high school students to learn physics or computer science. In recent years, multi-sensory experiential learning methods have begun to be incorporated into educational and school programs. Rather than having students learn only through traditional lecture-based methods, teachers are beginning to incorporate simulations, computer games, and more educational software items into their lesson plans. Computer simulations, games, and models can present difficult topics to students that may not be presentable in a traditional lecture-based format [26]. Research has also shown that combining traditional instruction with computer simulations on science teaching and learning is beneficial in promoting and developing students' content knowledge, science process skills, and coping skills with more complex tasks. For instance, Smetana and Bell [27] argue that computer simulations are successful in promoting learning because they allow students greater flexibility in exploring ideas and encourage them to justify their actions and answers in a timely manner. Review of the literature on modeling and simulation software and long-term learning outcomes suggests that simulation games in engineering education can promote the transferability of academic knowledge to industry. In other words, when an engineering simulation game is applied correctly, students will be more likely to sustain that knowledge and use it in their future in the industry [26]. These prior results suggest that the use of STEPP in the classroom has the potential to facilitate sustainable learning of physics and computational thinking—a supposition we tested after the present usability study was conducted.

### 1.4. Usability Testing and Educational Software

Part of the software development process involves the usability testing of early versions of the software with a small sample of target users. Developed in the 1980s, the primary function of usability testing is to examine the extent to which a product—in this case, software designed to increase physics learning and computational thinking—is easy

to use by the targeted users [28]. To assess usability, researchers observe the user experience (i.e., observe users sampled from the target audience as they learn to use the software) to determine the extent to which a software product can be used in the manner anticipated by the designers. This typically involves assessing the extent to which users find the product easy to use, whether users see the software as effective in facilitating the intended goals (i.e., in the case of educational software, whether it meets the learning goals), and identifying any design issues and/or bugs that interfere with usability of the program [29].

While there are many different ways to assess usability, the present research utilized the user-based method, a theoretical perspective on usability testing in which users' experiences using a software program are assessed via survey [28,30]. This theoretical perspective on usability testing typically assess the extent to which users' find a software program easy to use by asking participants to use the software and report on their experiences so that the researchers can identify problems users have with the software. As part of this process, user's actions and their overall satisfaction with the program are measured via survey. Because educational software varies by intended audience, the skills taught, and the overall subject matter, it is typical for usability studies to tailor the items they assess to the specific features of the software being tested.

Prior research on the usability of physics educational software has demonstrated the utility of usability testing and shown that these programs, when properly designed (i.e., are easy to use, intuitive, and focused on teaching specific concepts) can facilitate student learning [29,31]. While the majority of published usability studies have only assessed student responses after using educational software, there is limited evidence establishing the importance of assessing teacher feedback on educational software. The limited existing scholarship suggests that obtaining teacher buy in when introducing educational software in the classroom [32] as teachers have the final say in how such programs are utilized by their students [33].

To examine the usability of STEPP, we surveyed high school physics teachers who participated in a multi-day STEPP training workshop on their experiences using the software. This study also fills a gap in the literature in that usability studies in educational software largely examine students' perceptions, while neglecting the user experiences of teachers. The present study is one of the few studies to assess usability from the perspective of teachers—an often-overlooked group that is also important to study, as this group of users often selects the educational software that their students will use. In the case of STEPP, teachers were also expected to utilize the modules in class as part of their lectures. As a result, it is important to solicit feedback from this group of users regarding STEPP's ease of use, applicability to increasing student interest in STEM + Computing (STEM+C), and usefulness in the classroom.

### 1.5. Overview of the Present Study

The present study describes the results of our usability survey to determine whether the high school physics teachers attending the summer institute felt that the software was easy and enjoyable to use, would facilitate student physics and STEM-C learning, and would be useful in a high school physics classroom. These conceptual categories were selected based on the literature reviewed above on usability testing in educational software and the specific types of usability feedback needed during the software development process. Based on the prior research on usability testing of educational software reviewed above, we predicted that, relative to expectations, our participants would evaluate the modules more favorably across these measures after exposure to STEPP. We also sought feedback on any design issues that could hinder student ease of use, learning, or overall interest in potential STEM-C careers.

## 2. Materials and Methods

### 2.1. Participants and Design

A total of 12 high school physics teachers participated in an NSF-funded 4-day training institute held by the authors at a large southwestern university. The research team advertised this training institute to qualified physics high school teachers throughout the university's greater metropolitan area. Our usability study was incorporated into the planning of the institute, with teachers completing the time 1 survey at the beginning of the institute and time 2 at the end. Thus, the experimental design was a pre-posttest such that participants were asked to report their expectations for using STEPP, and then were asked to report on their actual experiences using STEPP.

Unfortunately, data from 4 participants were excluded owing to their failure to complete the pre- or post-test survey, reflecting a 33% attrition rate (i.e., 67% of the sample filled out both surveys) leaving us with a final sample of 8 teachers. The results below reflect input from the 8 teachers who completed both the time 1 and time 2 surveys.

The final sample consisted of 8 participants 8 (6 women, 2 men). They ranged in age from 21 to 47 (M = 33.8, SD = 7.8). Ethnicity was reported as follows: 4 (50%) White/Caucasian, 2 (25%) Asian, 1 (12.5%) Hispanic, and 1 (12.5%) Biracial/Multiracial. Their amount of time teaching Physics ranged as follows: 1 (12.5%) indicated 1–2 years, 2 (25%) indicated 3–5 years, 2 (25%) indicated 6–10 years, and 3 (37.5%) indicated 11–15 years.

The experimental design was a within subject design in which participants provided feedback on their expectations for using STEPP (time 1) and were once again sampled at the end of the institute (time 2).

### 2.2. Procedure

Participants were invited to apply for the institute via a flyer that described an opportunity to be on the cutting edge of teaching by learning the STEPP modules, which combined the power of Computer Science with Physics curriculum through a game engine. STEPP was described as a tool that could teach their students the basics of Newtonian mechanics through an interactive environment to build simulations. Only physics teachers who accurately followed our application instructions and were in the same metropolitan area as the university were selected for participation. This approach provided an educational opportunity to local teachers and promoted recruitment of these teachers for future field testing of our modules.

The research team received IRB approval to conduct a usability study as part of the teaching institute. Teachers enrolled in the institute received information on the usability study, and those willing to participate provided their informed consent. Consenting participants voluntarily completed the pre-test (time 1) questionnaire on the first day of the institute and the post-test (time 2) on the final day. In between the two assessments, participants learned to use the STEPP prototype modules. The institute was held for 7 h a day across the 4-day period. During this time, participants used STEPP for roughly 12 h.

### 2.3. Software

STEPP was designed to provide users with the opportunity to decompose word problems typically found in the teaching of introductory mechanics applicable to its three modules. First, a user selects a physics problem, such a standard word problem found in a textbook on the curriculum covered by the STEPP modules and levels. The motion of the object described in the problem is then decomposed into discrete states. The user then creates FSM states in STEPP corresponding to each state of motion and programs the transitions which cause the FSM to switch between these states. Inputs for the states and transitions vary by module and level. In module 1, level 1, users engage with a simple interface based on natural language. Subsequent levels and modules increase in complexity. User-programmed inputs include variables such as position, displacement, velocity, acceleration, and mass, with transitions defined by a user-determined final position, final time, or final velocity for each state. Incorrect input can create simulations

that fail to describe the desired motion, however, each simulation must be physically self-consistent in order to run. Motion in a new state must begin at the same time and position where a prior state ended, so the initial starting values of a new state are automatically created from the final values of the previous state. States can only be deleted in the reverse order in which they were created to prevent the introduction of unphysical errors in the programming of the simulation. If the states or transitions as programmed do not model a self-consistent physical system, the STEPP simulation will not run and an error message is returned to the user with information on how to correct the issue. Thus, students are provided a scaffolded environment in which they can safely take a trial-and-error approach to problem solving. By using scaffolded FSMs to program representations of word problems encountered in the context of the physics classroom, students and teachers can engage in creation of their own simulations of motion without risk of either software failure or the accidental modeling of unphysical situations.

The STEPP environment includes sub-windows and graphic icons and diagrams. The diagram at the bottom of the main window is the Finite State Diagram (i.e., Machine) window which controls the physical motion. Each discrete higher-level state of the FSM is a partition of a lower-level state space characterized by continuous values of time, displacement, velocity, and acceleration. This architecture is hybrid in that continuous (e.g., lower-level state space) and discrete (higher-level state space) are combined and integrated. An example task would be a ball moving up and then down. When the ball moves up, this can be considered one higher-level discrete state labeled "up". Within "up", there are lower-level partitions of the continuous space corresponding to motion that matches the quantitative definition of "up", i.e., positive vertical motion. Similar reasoning can be applied for the state "down." See Figure 2 for screenshots of the final product, informed by the results of the present study, which capture the STEPP modules' scaffolding as well as its multiple representations, color-coded actions, and animations.

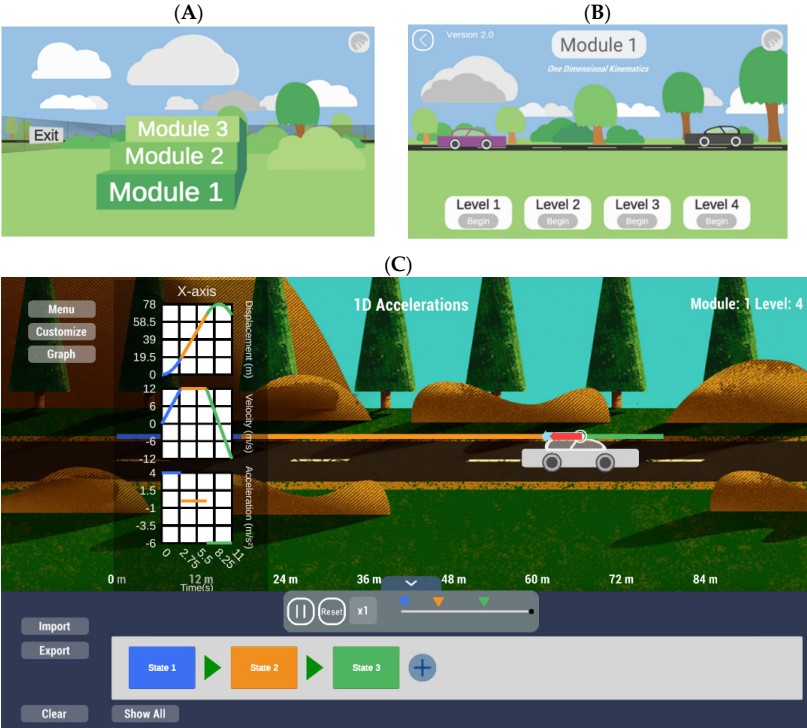

**Figure 2.** Screenshots of the current version of STEPP depicting (**A**) the STEPP Module-selection menu; (**B**) the STEPP level-selection menu within Module 1; and (**C**) STEPP Module 1, Level 4 focused on 1D acceleration as improved by the results of this usability study of high school physics teachers.

For an example of a concrete task solved via STEPP, we turn to "Exercise 2.43 Launch Failure" in a typical introductory physics textbook [34]:

"A 7500-kg rocket blasts off vertically from the launch pad with a constant upward acceleration of 2.25 m/s$^2$. and feels no appreciable air resistance. When it has reached a height of 525 m, its engines suddenly fail; the only force acting on it now is gravity.

(a)  What is the maximum height this rocket will reach above the launch pad?
(b)  How much time will elapse after engine failure before the rocket comes crashing down to the launch pad, and how fast will it be moving just before it crashes?
(c)  Sketch $a_y$-t, $v_y$-t, and y-t graphs of the rocket's motion from the instant of blast-off to the instant just before it strikes the launch pad".

We can solve this problem by modeling it as a hybrid-level finite-state machine within STEPP. We can describe the evolution of this physical system be decomposing it into three high-level states: (1) the rocket rises with constant upward acceleration before its engines fail, (2) the rocket continues to ascend until it comes to rest at the apex of its flight, and (3) the rocket falls back to Earth and crashes on the launch pad. Each of these discrete high-level states can be further partitioned into the low-level variables of time, position, velocity, and acceleration that evolve continuously between the transitions at the start and end of each high-level state.

State 1 "Powered ascent" is characterized by a constant upward acceleration of 2.25 m/s$^2$. It begins with the rocket launch at $t_{1i}$ = 0 s, $y_{1i}$ = 0 m, $v_{1i}$ = 0 m/s, where the number of the subscript corresponds to the high-level state and the letter ("i" or "f") corresponds to the initial or final value at the transition that begins/ends the high-level state. State 1 ends at the transition corresponding to engine failure (at a height of $y_{1f}$ = 525 m). State 2 "Unpowered ascent" is characterized by an upward (positive) velocity and gravitational acceleration of g = −9.8 m/s$^2$. It begins with the same values of the low-level variables as state 1 ends (no "time travel" -> time is continuous, no "teleportation" -> y is continuous, conservation of linear momentum -> v is continuous). State 2 ends at the transition corresponding to rest at the apex of the flight ($v_{2f}$ = 0 m/s). State 3 "Descent" is characterized by a downward (negative) velocity and gravitational acceleration of g = −9.8 m/s$^2$. It begins at the end of state 2 and ends when the rocket crashes on the launch pad ($x_{3f}$ = 0 m).

Once the student has decomposed the problem into these three high-level states and determined the values of the low-level variables that describe the transitions, they can program STEPP to produce a simulation of the rocket's flight, with an animation and real-time graphing providing feedback in multiple representations. STEPP also provides warnings if the student programs unphysical transitions. For example, if the student had forgotten that the gravitational acceleration was negative and had incorrectly inputted g = +9.8 m/s$^2$ for state 2, the rocket would accelerate upwards and never come to rest. In this case, the transition $v_{2f}$ = 0 m/s could not be realized and the simulation would never end. The current version of STEPP provides the error message "You'll never reach that velocity. The acceleration and change in velocity do not agree." This message is intended to guide the student to correct their mistake without becoming discouraged. Figure 3 shows a screen capture from the end of the STEPP simulation. States 1, 2, and 3 are shown in blue, orange, and green, respectively, with state 1 opened to show the low-level variables describing its initial and final transitions. By opening the other states or reading from directly from the graphs, the student can solve the problem: (a) the rocket reaches a maximum height of 646 m at the end of state 2/beginning of state 3 (orange/green peak in top graph), (b) the rocket comes crashing down to the launch pad $t_{3f}$–$t_{1f}$ = 16.4 s after engine failure at a final velocity of −113 m/s (right end of the green line segment in middle graph), and (c) STEPP has conveniently provided us with the desired graphs.

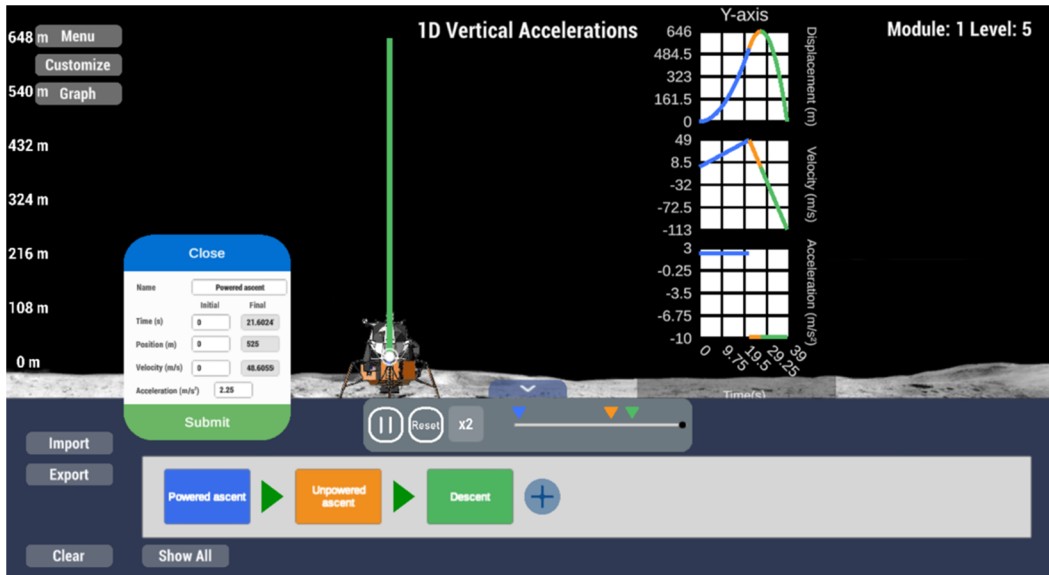

**Figure 3.** Screenshot of the end of the simulation describing the rocket flight in "Exercise 2.43 Launch Failure" [34]. The animation shows the rocket's flight, depicted in STEPP as the Apollo lunar module rising and falling from the Moon's surface. The state diagram shows the three high-level states: state 1 "Powered ascent", state 2 "Unpowered ascent", and state 3 "Descent" (shown in blue, orange, and green, respectively). The graphs show how the low-level variables (position, velocity, and acceleration) evolve with time, color-coded by the high-level state. The open panel shows the acceleration and initial and final time, position, and velocity of state 1 "Powered ascent".

### 2.4. Measures

The authors developed a short survey with 13 continuous items assessing participants attitudes toward STEPP modules across three different conceptual categories assessed used in usability studies and relevant to our specific software tool: User Experience (e.g., "I like using STEPP"), STEM-C relevance (e.g., "STEPP is useful for teaching Physics"), and Classroom Applicability (e.g., STEPP is valuable for students learning physics") on Likert scales ranging from 1 = "strongly disagree" to 7 = "strongly agree" with 4 as the midpoint labeled "neutral.". See Figure 4 for abbreviated descriptions of each item and Table 1 below for a copy of the full scale along with the observed correlations between time 1 and time 2.

To collect more nuanced feedback on our software, we also asked participants to provide written responses to several open-ended questions that asked participants to provide more detail regarding their impressions of STEPP. These items allowed participants to provide detailed feedback on what they liked and did not like about STEPP, and what they found easy to use and thought could help them teach physics. At time one, these items assessed participants expectations about using STEPP and were phrased to indicate that we were interested in understanding their expectations. At time two, these items assessed participants' actual experiences after using STEPP throughout the teacher training.

Data were collected via Qualtrics survey software. At time 1, we asked a total of 13 quantitative questions and 5 opened-ended questions. At time 2, we re-assessed the same 13 quantitative items and added 2 additional open-ended responses to gather additional feedback on recommended changes after participants had used STEPP.

**Table 1.** The Quantitative items used to assess teacher's usability experience with STEPP along with the correlation between time 1 and time 2 for each pair of questions. Time 1 wording appears in brackets.

| Item | Conceptual Category | T1 vs. T2 Correlation |
|---|---|---|
| 1. STEPP [will be] was easy to use. | User Exp | 0.54 |
| 2. I [will like] liked the look of STEPP. | User Exp | −0.20 |
| 3. I [will like] liked using STEPP. | User Exp | 0.28 |
| 4. [I expect that] Using STEPP increased my interest in physics. | STEM+C | 0.61 |
| 5. [I expect that] Using STEPP increased my interest in computer science. | STEM+C | 0.44 |
| 6. [I expect that] Using STEPP increased my interest in STEM (Science Technology Engineering Mathematics). | STEM+C | 0.37 |
| 7. [I expect that after using it,] I want to use STEPP more. | User Exp | 0.41 |
| 8. [I expect that] Using STEPP will help me teach physics. | STEM+C | 0.90 ** |
| 9. [I expect that] Using STEPP made me feel more confident that I use it in my classroom to teach physics. | User Exp | 0.72 * |
| 10. I would like to use STEPP in my classroom to help my students learn physics. | CLASS APP | 0.83 * |
| 11. [I expect that] STEPP is valuable for students learning physics. | CLASS APP | 0.91 ** |
| 12. [I expect that] The state diagram is useful for breaking down physics problems I would use with my class into discrete steps. | CLASS APP | 1.0 *** |
| 13. [I expect that] Concept maps are useful for breaking down physics problems I would use with my class into discrete steps. | CLASS APP | 0.66 |

Notes: all items were assessed on a continuous scale ranging from 1 ("strongly disagree") to 7 ("strongly agree") with a midpoint of 4 ("neutral"). Phrases above in brackets provide the wording for the pre-test survey. Statistically significant correlations are denoted as follows: * $p < 0.05$; ** $p < 0.01$; *** $p < 0.001$.

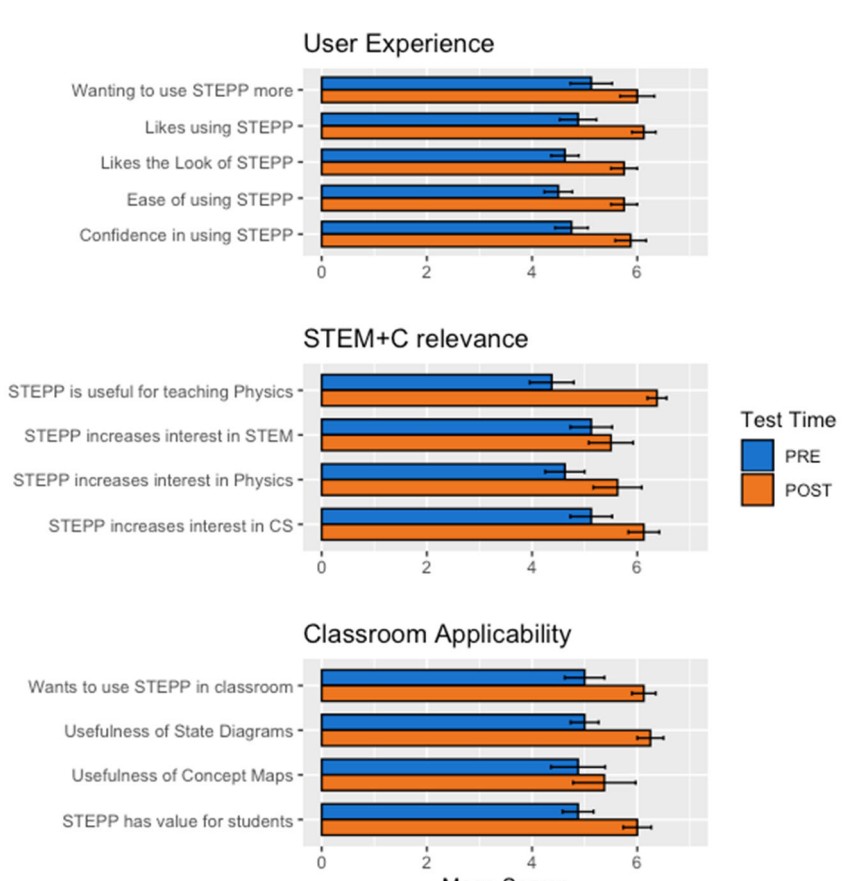

**Figure 4.** Mean evaluations at pre- and post-test on the quantitative items.

## 3. Results

We analyzed changes in attitude in three conceptual topics: (1) User Experience, (2) STEM-C Relevance, and (3) Classroom Applicability. To compare changes in teachers' attitude before and after using STEPP, we used within subjects' *t*-tests and report the means, standard deviations, mean difference, correlations between time 1 and time 2 measures, *t*-test statistic, Cohen's d effect size, *p*-value, and observed statistical power for each analysis reported below. See Figure 3 for a graphical display of pre-post means for all measures. Note that while each item was analyzed individually but the reliability for the combined items for each the conceptual topic at both Time 1 and Time 2 are reported below. The correlations between pre- and post-test items is reported in Table 1 above.

An anonymous reviewer asked us to reanalyze the data using a non-parametic equivalent of the within subjects' *t*-test, the Wilcoxon Signed Rank Test. We re-ran the analyses and did not find our results changed for any of the items. Also note that the inclusions of scale reliabilityies for the three conceptual categories was based on similar reviewer feedback.

### 3.1. User Experience (Time 1 Chronbach's α = 0.88; Time 2 Chronbach's α = 0.57)

After using STEPP, teachers found it easier to use than expected, (Time 1 M = 4.5, SD = 0.33; Time 2 M = 5.75, SD = 0.25, D = −1.25), $t(7) = 5$, d = 1.71, $p = 0.002$, Obs 1−β = 0.56. Teachers also liked the look of the modules more than they expected, (Time 1 M = 5.125, SD = 1.61; Time 2 M = 6, SD = 0.33; D = −0.875), $t(7) = 2.83$, d = 1.55, $p = 0.026$, Obs 1−β = 0.91. STEPP exceeded teacher's pre-test expectations on how much they would like using the modules, (Time 1 M = 4.875, SD = 0.96; Time 2 M = 6.125, SD = 0.17; D = −1.25), $t(7) = 3.03$, d = 1.5, $p = 0.019$, Obs 1−β = 0.85. There were no significant differences, however, between the degree to which teachers reported wanting to use STEPP and to their expectations of STEPP, (Time 1 M = 5.125, SD = 1.61; Time 2 M = 6, SD = 0.73; D = −0.875), $t(7) = 2.2$, d = 0.84, $p = 0.064$, Obs 1−β = 0.59. Teachers became confident in using the STEPP modules more than they expected, (Time 1 M = 4.75, SD = 0.62; Time 2 M = 5.875, SD = 0.49; D = −1.125), $t(7) = 4.97$, d = 1.3, $p = 0.002$, Obs 1−β = 0.28.

### 3.2. STEM-C Relevance (Time 1 Chronbach's α = 0.79; Time 2 Chronbach's α = 0.77)

After use, teachers judged that STEPP would increase interest in Physics more than they expected, (Time 1 M = 4.625, SD = 1.27; Time 2 M = 5.625, SD = 2.88; D = −1), $t(7) = 2.65$, d = 0.83, $p = 0.033$, Obs 1−β = 0.44. Teachers' judgement of how much STEPP would increase interest in Computer Science also exceeded their expectation prior to use, (Time 1 M = 5.125, SD = 1.61; Time 2 M = 6.125, SD = 0.48; D = −1), $t(7) = 2.65$, d = 0.99, $p = 0.033$, Obs 1−β = 0.58. Teachers did not differ in how much they believed STEPP could increase interest in STEM fields before and after using the modules, (Time 1 M = 5.125, SD = 1.61; Time 2 M = 5.5, SD = 2.04; D = −0.375), $t(7) = 0.81$, d = 0.32, $p = 0.442$, Obs 1−β = 0.93. Teachers did, however, report a strong change in their appreciation for STEPP as a tool for teaching Physics, (Time 1 M = 4.375, SD = 1.99; Time 2 M = 6.375, SD = 0.07; D = −2), $t(7) = 7.48$, d = 1.18, $p < 0.001$, Obs 1−β = 0.14.

### 3.3. Classroom Applicability (Time 1 Chronbach's α = 0.83; Time 2 Chronbach's α = 0.75)

After use, teachers reported wanting to use STEPP in the classroom at a higher rate after use, (Time 1 M = 5, SD = 1.31; Time 2 M = 6.125, SD = 0.17; D = −1.125), $t(7) = 4.97$, d = 1.01, $p = 0.002$, Obs 1−β = 0.13. Teachers evaluated STEPP to have a higher value to students after having used the program themselves, (Time 1 M = 4.875, SD = 0.48; Time 2 M = 6, SD = 0.33; D = −1.125), $t(7) = 9$, d = 1.38, $p < 0.001$, Obs 1−β = 0.23. Concept Maps [27] were used in the summer institute/curriculum design as a bridge from a more common classroom tool to the unfamiliar State Diagrams. Teachers did not find the Concept Maps in STEPP to be useful in breaking down physics more after using STEPP, (Time 1 M = 4.875, SD = 4.51; Time 2 M = 5.375, SD = 8.06; D = −0.5), $t(7) = 1.08$, d = 0.31, $p = 0.316$, Obs 1−β = 0.46. However, there was a large difference between how useful teachers found the State Diagrams in STEPP after using the modules relative to before

use, (Time 1 M = 5, SD = 0.33; Time 2 M = 6.25, SD = 0.25; D = −1.25), $t(7)$ = 3.42, d = 1.71, $p$ = 0.011, Obs 1−β = 0.87.

### 3.4. Qualitative Feedback

Participants generally reported that they expected STEPP to be more coding-focused and difficult to use but found instead that it was more physics focused instead. As one participant reported: "I initially envisioned more of a direct coding component, but this was not present in STEPP as STEPP instead introduced CS conceptually through state diagrams".

In terms of what participants liked most about STEPP, their responses uniformly focused on the ability of STEPP to break problems down into discrete, logical steps. Participants unanimously reported minor bugs as the issue they liked least about STEPP. Similarly, they all focused on how the ability to break problems down into discrete steps would be the biggest benefit of using STEPP to teach physics in their classrooms. When asked about how to make STEPP easier to use, participants focused on more detailed instructions. For instance, one participant stated: "Provide more user guidance (maybe through a tutorial at first use?)" When asked about STEPP's usefulness in teaching computational thinking to students, the teachers also focused on the discrete steps approach to problem solving as the modules' best contributing factor.

For the post-test only items, participants suggested including more instructions in STEPP. The final question asked for any additional input and their feedback split into two equal categories: an expression of enthusiasm for using it in their classrooms and a desire for more teaching resources to help support them as they implement the modules into their classes. As one participant expressed: "Keep on keepin' on! This is promising software".

## 4. Discussion

Overall, our results supported our predictions in that teachers' expectations for STEPP were exceeded after using it. Furthermore, their overall feedback suggests that these software modules are easy and enjoyable to use, can facilitate interest in and learning of physics and STEM-C, and teachers see the utility of these modules for use in the classroom. This was illustrated by both the quantitative and qualitative responses from our sample of teachers. The qualitative items in particular also captured important user feedback and suggestions which may be capitalized upon in the further development of the STEPP modules. Thus, the user feedback we received was not only consistent with our predictions, but also helpful for the future development of STEPP, which may include bugs fixes and adding a tutorial mode.

These results further support the notion that, as with other types of educational physics software [27,29,30,35], STEPP has the potential to facilitate student understanding of modeling, simulation, and the transitions between different states. These concepts are also useful for understanding the impact of sustainability in physics education. Furthermore, this study adds to the growing literature on the usability of physics educational software by assessing usability from the perspective of teachers—an often overlooked group. This is also important because of teachers' role in selecting the educational software used by both students and teachers inside and outside the classroom. Thus, our results also highlight the importance of the teacher's perspective on educational software in the classroom.

While the research team found the results of the present study promising, this research is not without its limitations. For instance, while we assessed usability and perceived relevance to STEM-C, we did not assess learning. This is because (a) it would make little sense to assess the sample of physics teachers in this manner in the context of a 4-day summer institute; (b) we planned to test student learning after completing our usability studies. Furthermore, the present study has a small sample size—an issue typically problematic in research. However, this is not unusual for usability research where small samples are the norm. Illustrative of this, research indicates that 80% of usability problems are detected with samples as small as 4–5 participants [36]. We intentionally designed a within-subjects usability test as this type of design is also known to accommodate small

sample sizes [37]. Based on the generally large effects [38] and high statistical power observed in our results, we maintain that our sample size was adequate for our purposes.

It is notable that we did not conduct multivariate analyses because some of the scaled reliabilities decreased at time 2. This may reflect the notion that the teacher's expectations were more conceptually similar while their hands-on experience with STEPP produced more variability in their post-test responses. Finally, we need to acknowledge the limited generalizability of this research, as these results are specific to our specific software and may not be applicable to other types of users and other types of educational software.

In the future, our research will examine the usability of our software from the perspective of students as well. However, this initial test provides some evidence that STEPP may be an easy tool to increase student interest in and commitment to STEM careers.

## 5. Conclusions

Overall, this research addresses a gap in the literature by examining usability from the perspective of teachers. Furthermore, this work suggests that simulations may have a promising role in the future of physics education, modeling, simulation, computational thinking, and the sustainability of systems. While people may generally think of sustainability as referring to some ecological result, such as zero energy or agriculture. Thus, the present research applies sustainability to education by considering the role that educational software plays in long-term learning outcomes. We posit that multiple representations and scaffolding are two educational principles which lead to a more diverse and robust system (STEPP). Having these two principles in action may make STEPP more accessible with broad impact.

**Supplementary Materials:** More details on STEPP are available online at https://stepp.utdallas.edu.

**Author Contributions:** Conceptualization, R.E.G., V.G., P.F., M.K. (Midori Kitagawa), M.U. and M.K. (Michael Kesden); methodology, R.E.G. and V.G.; software, P.F., M.K. (Midori Kitagawa), M.K. (Michael Kesden), C.K., N.T., R.J. and A.R.; data collection, R.E.G., V.G., M.U., K.S. and B.H.; data analyses, R.E.G. and V.G.; project administration, M.K. (Midori Kitagawa), K.S. and C.K.; writing—original draft preparation, R.E.G., V.G., H.P., P.F., M.U., M.K. (Midori Kitagawa) and M.K. (Michael Kesden); writing—review and editing, H.P., M.K. (Midori Kitagawa), P.F., M.U. and M.K. (Michael Kesden); funding acquisition, M.K. (Midori Kitagawa), P.F., M.K. (Michael Kesden), M.U. and R.E.G. All authors have read and agreed to the published version of the manuscript.

**Funding:** This research was funded by the National Science Foundation grant number DRL-1741756.

**Institutional Review Board Statement:** The study was conducted according to the guidelines of the Declaration of Helsinki, and approved by the Institutional Review Board of the University of Texas at Dallas (protocol # 19MR0153, approved 19 June 2019).

**Informed Consent Statement:** Informed consent was obtained from all subjects involved in the study.

**Data Availability Statement:** The data reported is this manuscript will be made available on our website.

**Acknowledgments:** The authors wish to thank the high school teachers who participated in this study and the many students who helped build this software.

**Conflicts of Interest:** The authors declare no conflict of interest.

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
