# Peer review of "A Usability Study of Classical Mechanics Education Based on Hybrid Modeling: Implications for Sustainability in Learning"

_sustainability, doi:10.3390/su132011225_

Round 1
Reviewer 1 Report
Thank you for the opportunity to read this interesting work.
There is some 'sliding' in the Introduction between physics education and computer science education. The paper is about a computer-based tool for physics education, so I would argue that the statistics and citations about computer science education are irrelevant and should be removed. The notion that the tool teaches both physics content and computational thinking is not tested.
The paper as a whole needs a more clear and coherent 'story line', or some subheadings. At the moment it moves from one topic to another in a way that disorients the reader.
Sustainability is mentioned in the paper, presumably in a nod to the title of the target journal, but I'm not sure that it is used in a way compatible with the intentions of the journal. I think, rather than 'shoe-horn' in a reference that doesn't really work, it might be better to remove the reference: I think there is enough of a case for the usefulness of the work without it.
This sentence in the Discussion, and the whole of the Conclusion, makes claims that are not in evidence in the useability data reported in this paper: "These results further support the notion that STEPP has the potential to facilitate stu-dent understanding of modeling, simulation, and the transitions between different states. These concepts are also useful for understaninding the impact of sustainability in phyics.". (note too the two typos in the last sentence) No eidence is adduced at all about student understanding. The paper presents data on teachers' impressions of useabilility. No paper should make claims not present in evidence. The only claims that can be made in this paper are about teachers' impressions of useability.
The paper does not mention approval of the study by a Research Ethics Board/Institutional Review Board, nor free and informed consent by participants: this is important information to include in a research paper.
The project of creating, testing and using STEPP is an interesting and worthwhile one, and the paper has potential, but it needs attention to structure and editing, removal of excessive claims not supported by the evidence and addition of information about research ethics before it is ready for publication.
Author Response
Comments raised by Reviewer 1:
- Reviewer 1 said: “The paper is about a computer-based tool for physics education, so I would argue that the statistics and citations about computer science education are irrelevant and should be removed.”
- Thank you for this feedback. In our brevity, it appears that we did not emphasize that the goal of this physics tool was to increase both physics knowledge as well as Computational Thinking (a core characteristic needed for computer scientists). We have further emphasized this link. If, upon reviewing the revised sessions, it would be preferrable to remove the section, please let us know – we would be happy to do this.
- Reviewer 1 stated: “The paper as a whole needs a more clear and coherent 'story line', or some subheadings.”
- Thank you for this feedback. We have added subheadings to the introduction and we also expanded sections of the manuscript to improve its flow.
- Reviewer 1 recommended we remove the reference of sustainability in the paper.
- Thank you for this feedback. Given that the name of the journal is Sustainability, and this is a special issue on modeling and sustainability, this request may not be feasible. With further detail from Reviewer 1, we would be happy to address the specific issue caused by our use of the term. Please advise.
- Reviewer 1 mentions two typos in this sentence that should be fixed: "These results further support the notion that STEPP has the potential to facilitate student understanding of modeling, simulation, and the transitions between different states. These concepts are also useful for understaninding the impact of sustainability in phyics.".
- Thank you for this feedback. Both typos have been corrected.
- Reviewer 1 said: “No evidence is adduced at all about student understanding. The paper presents data on teachers' impressions of usability. No paper should make claims not present in evidence. The only claims that can be made in this paper are about teachers' impressions of useability.”
- Thank you for this feedback. Any comments regarding the hypothesized impact of STEPP on physics learning and computational thinking have been edited in accordance with this feedback.
- Reviewer stated: mention of IRB approval and informed consent should be included in the paper.
- Thank you for this feedback. We have edited the procedure to clearly indicate that we received IRB approval to conduct this usability study and that participants provided informed consent to participate in this usability study. See page 4.
- Reviewer 1 said: “(The paper) needs attention to structure and editing, removal of excessive claims not supported by the evidence and addition of information about research ethics before it is ready for publication.”
- Thank you for this feedback. By addressing the helpful, detailed, and specific feedback from Reviewer 1, we believe we have addressed this global concern. However, if there are additional issues we did not address, we would be happy to do so with further guidance.
Reviewer 2 Report
1. Do not present relevant results in the abstract.
2. The analysis sample is quite limited. Only 8 teachers. It does not clearly justify the rationale for them.
3. The validation of the analysis criteria within the methodological process is unknown. How were the pre-test and post-test carried out? How was it selected?
4. It presents a starting hypothesis that is not clearly corroborated in the data obtained.
5. It does not explain in detail the results obtained.
6. Extend the conclusions. Too brief. Is the initial hypothesis fulfilled?
Author Response
- Reviewer 2 said: “Do not present relevant results in the abstract.”
- When reviewing the editor’s comments, it appears that the editor said the opposite. As a result, this portion of the abstract was not changed. Please provide clarification on how we should change the text of the abstract and we will happily do so.
- Reviewer 2 said: “The analysis sample is quite limited. Only 8 teachers. It does not clearly justify the rationale for them.”
- Thank you for this feedback. This issue was also raised by the Editor. We have now clarified in the manuscript that small samples are the norm in usability research.
- Reviewer 2 stated: “The validation of the analysis criteria within the methodological process is unknown. How were the pre-test and post-test carried out? How was it selected?”
- Thank you for this feedback. While we do not understand what the Reviewer meant by “The validation of the analysis criteria within the methodological process is unknown”, we would be more than happy to address this concern if the reviewer were to provide us more clarity. While we did state in the prior version of the manuscript that the items were author-generated and assessed via Qualtrics, we added a paragraph explaining more in detail our rationale for assessing these items.
- Reviewer 2 said: “It presents a starting hypothesis that is not clearly corroborated in the data obtained.”
- Thank you for this feedback. We have edited the introduction and discussion of the manuscript to make it clear that our usability study did not assess whether our software increases physics knowledge and computational thinking. This hypothesis was tested in a separate study conducted after we assessed the usability of the software. We make this distinction clearer in our revised manuscript. See pages x and y.
- Reviewer 2 stated: “It does not explain in detail the results obtained.”
- Thank you for this feedback. While we do not understand what the Reviewer meant by this comment. We would be happy to revise with further clarification.
- Reviewer 2 said: “Extend the conclusions. Too brief. Is the initial hypothesis fulfilled?”
- Thank you for this feedback. We have expanded the discussion accordingly.
Reviewer 3 Report
Shortcomings and suggestions --------------------------------------------------------------
1. Most references (e.g. [1-10]) are simply outdated. Please replace them by those published in the last 5-10 years.
2. Please add subsection “Software” in section “Materials and Methods” to describe, in sufficient details, the use of the STEPP environment (decomposing a system/task in key variables, linking them though states, avoiding undesirable transitions, etc.). In other words, by using a concrete task, explain the main activities in the hybrid modelling applied.
3. It would be useful to defend the three dimensions applied by using a theoretical framework that refers to them.
4. Concerning the three measures applied, please reports their (Cronbach’s alpha) reliability.
5. Please report just the overall means (or medians if non-parametric statistics should be applied) of the three measures, and examine their pretest-posttest differences using appropriate statistical tests (probably rather non-paramatric than parametric). Please do not consider individual indicators (e.g., “Likes using STEPP”) unless the examined individual measures are of satisfactory reliability as well (to determine their reliability, the formula for correction for attenuation may be applied).
6. Please clarify how many days and how many hours each day the teachers work with the STEPP environment.
7. Please examine the results of this study within a relevant research context, referring to the findings of other studies published recently. List these studies in the Discussion part (no such study at present; at least 3-4 studies should be added in this part).
8. It seems that this study involved 12 participants. If so, please mention this. As the size of the sample is small (please acknowledge this fact when the limitations of the study are considered), please do not use percentages in subsection 2.1; it is enough to say two (2) participants were Asian, etc.
--------------------------------------------------------------- End --------------------
Author Response
Comments raised by Reviewer 3:
- Reviewer 3 said: “Most references (e.g. [1-10]) are simply outdated. Please replace them by those published in the last 5-10 years.”
- Thank you for this feedback. When able to find suitable citations that were newer, we swapped them out as recommended. Not all citations were replaceable, however. As a result, we would hope that the reviewer and editor recall that science tends to be additive, and knowledge builds over time so a few older citations may be a reflection of this.
- Reviewer 3 stated: “Please add subsection “Software” in section “Materials and Methods” to describe, in sufficient details, the use of the STEPP environment (decomposing a system/task in key variables, linking them though states, avoiding undesirable transitions, etc.). In other words, by using a concrete task, explain the main activities in the hybrid modelling applied.”
- Thank you for this feedback. We have added this section.
- Reviewer 3 mentioned that: “It would be useful to defend the three dimensions applied by using a theoretical framework that refers to them.”
- Thank you for this feedback. We have added this information in the section on prior usability testing.
- Reviewer 3 said: “Concerning the three measures applied, please report their (Cronbach’s alpha) reliability.”
- Thank you for this feedback. We have added this information. Note that the three dimensions were conceptual rather than statistical, so they were intended to group our results rather than to build reliable scales. We wanted to understand the more nuanced feedback provided by the individual items.
- Reviewer 3 stated: “Please report just the overall means (or medians if non-parametric statistics should be applied) of the three measures and examine their pretest-posttest differences using appropriate statistical tests (probably rather non-paramatric than parametric).
- Thank you for this feedback. We have added this information to the results section.
- Reviewer 3 stated: “Please do not consider individual indicators (e.g., “Likes using STEPP”) unless the examined individual measures are of satisfactory reliability as well (to determine their reliability, the formula for correction for attenuation may be applied).”
- Thank you for this feedback. This feedback is unclear to us. Please let us know what exactly is meant by this comment and we will be happy to address this.
- Reviewer 3 said: “Please clarify how many days and how many hours each day the teachers work with the STEPP environment.”
- Thank you for this feedback. We have added this information to the methods section.
- Reviewer 3 said: “Please examine the results of this study within a relevant research context, referring to the findings of other studies published recently. List these studies in the Discussion part (no such study at present; at least 3-4 studies should be added in this part).”
- Thank you for this feedback. We have added citations to other usability studies of software intended to enhance learning to the introduction.
- Reviewer 3 stated: “It seems that this study involved 12 participants. If so, please mention this. As the size of the sample is small (please acknowledge this fact when the limitations of the study are considered), please do not use percentages in subsection 2.1; it is enough to say two (2) participants were Asian, etc.”
- Thank you for this feedback. We have acknowledged the small sample size as a limitation of the present work and also clarified how we ended up with a sample of 8 when 12 teachers participated in the teacher training. We have adjusted the percentages in subsection 2.1 as requested.
Round 2
Reviewer 1 Report
Thank you for the attention you have paid to the recommendations made and issues raised in the review process. In my view the paper is much improved and now ready to be accepted for publication.
Author Response
Thank you for this feedback. Your input has improved our research.
Reviewer 2 Report
In my opinion, the manuscript is ready for publication.Author Response
Thank you for this feedback. Your input has improved our research.
Reviewer 3 Report
Although the authors improved their contribution, they do not adequately address a number of suggestions (see below). Hence, the scientific value of this contribution is rather low and it doesn’t deserve to be published in an outstanding research journal such as Sustainability. Had the authors examined the three dimensions (not their individual indicators) within a suitable theoretical framework and discussed the results concerning the dimensions in the context of other (relevant and recently published) papers, they would probably increase the research significance of their contribution, making it more appropriate for publication in the Journal.
Final Ratings: Between Low and Average
----------------
Reviewer 3 stated: “Please add subsection “Software” in section “Materials and Methods” to describe, in sufficient details, the use of the STEPP environment (decomposing a system/task in key variables, linking them though states, avoiding undesirable transitions, etc.). In other words, by using a concrete task, explain the main activities in the hybrid modelling applied.”
Authors’ comments: Thank you for this feedback. We have added this section.
Reviewer’ final comment: The authors didn’t use a concrete task and explain the use of STEPP to solve it.
----------------
Reviewer 3 mentioned that: “It would be useful to defend the three dimensions applied by using a theoretical framework that refers to them.”
Authors’ comments: Thank you for this feedback. We have added this information in the section on prior usability testing.
Reviewer’ final comment: The authors didn’t use any framework that refers to these dimensions.
----------------
Reviewer 3 stated: “Please report just the overall means (or medians if non-parametric statistics should be applied) of the three measures and examine their pretest-posttest differences using appropriate statistical tests (probably rather non-paramatric than parametric).
Authors’ comments: Thank you for this feedback. We have added this information to the results section.
Reviewer’ final comment: Due to a small sample, non-parametric (not parametric) statistics needs to be applied.
----------------
Reviewer 3 stated: “Please do not consider individual indicators (e.g., “Likes using STEPP”) unless the examined individual measures are of satisfactory reliability as well (to determine their reliability, the formula for correction for attenuation may be applied).”
Authors’ comments: Thank you for this feedback. This feedback is unclear to us. Please let us know what exactly is meant by this comment and we will be happy to address this.
Reviewer' final comment: Had the authors searched the Internet using keywords “reliability” and “correction for attenuation”, they would easily realize the context of this suggestion. Unless the examined individual measures are of satisfactory reliability (e.g., at least 0.70), the use of individual indicators (e.g., “Likes using STEPP”) is not appropriate. The authors didn't report the reliability of the individual measures.
----------------
Reviewer 3 stated: “It seems that this study involved 12 participants. If so, please mention this. As the size of the sample is small (please acknowledge this fact when the limitations of the study are considered), please do not use percentages in subsection 2.1; it is enough to say two (2) participants were Asian, etc.”
Authors’ comments: Thank you for this feedback. We have acknowledged the small sample size as a limitation of the present work and also clarified how we ended up with a sample of 8 when 12 teachers participated in the teacher training. We have adjusted the percentages in subsection 2.1 as requested.
Reviewer's final comment: There is no sense to use percentages for a whole comprising 8 units (in this case 8 participants).
----------------
Reviewer 3 said: “Please examine the results of this study within a relevant research context, referring to the findings of other studies published recently. List these studies in the Discussion part (no such study at present; at least 3-4 studies should be added in this part).”
Authors’ comments: Thank you for this feedback. We have added citations to other usability studies of software intended to enhance learning to the introduction
Reviewer 3 comment (the most critical issue): This is not done. The authors only used references related to the sample applied. Without this examination in the Discussion part (a must for empirical research!), it is hard to estimate to what extent, if any, this contribution may advance the field.
---------------- End of review
Author Response
Comments raised by Reviewer 3:
- Reviewer 3 stated: “Although the authors improved their contribution, they do not adequately address a number of suggestions (see below). Hence, the scientific value of this contribution is rather low and it doesn’t deserve to be published in an outstanding research journal such as Sustainability. Had the authors examined the three dimensions (not their individual indicators) within a suitable theoretical framework and discussed the results concerning the dimensions in the context of other (relevant and recently published) papers, they would probably increase the research significance of their contribution, making it more appropriate for publication in the Journal.”
- Thank you for this feedback. While we have addressed the specific outstanding concerns listed below, we believe there is a fundamental misunderstanding about our research plan and have changed attempted to clarify accordingly. For instance, the word “dimension” to “conceptual categories” in the manuscript to clarify that the 13-item scale was developed to examine 13 unique/individual questions that generally fit into 3 conceptual categories. It was never our intention to create 3 subscales mathematically and this does not make sense in the content of our research and in usability testing more broadly.
- Reviewer 3 stated: “The authors didn’t use a concrete task and explain the use of STEPP to solve it.”
- Thank you for this feedback. We have now added the following text with accompanying figure (see Figure 3) in the revised manuscript:
“For an example of a concrete task solved via STEPP, we turn to “Exercise 2.43 Launch Failure” in a typical introductory physics textbook [1]:
‘A 7500-kg rocket blasts off vertically from the launch pad with a constant upward acceleration of 2.25 m/s2. and feels no appreciable air resistance. When it has reached a height of 525 m, its engines suddenly fail; the only force acting on it now is gravity.
What is the maximum height this rocket will reach above the launch pad?
How much time will elapse after engine failure before the rocket comes crashing down to the launch pad, and how fast will it be moving just before it crashes?
Sketch ay-t, vy-t, and y-t graphs of the rocket’s motion from the instant of blast-off to the instant just before it strikes the launch pad.’
We can solve this problem by modeling it as a hybrid-level finite-state machine within STEPP. We can describe the evolution of this physical system be decomposing it into three high-level states: (1) the rocket rises with constant upward acceleration before its engines fail, (2) the rocket continues to ascend until it comes to rest at the apex of its flight, and (3) the rocket falls back to Earth and crashes on the launch pad. Each of these discrete high-level states can be further partitioned into the low-level variables of time, position, velocity, and acceleration that evolve continuously between the transitions at the start and end of each high-level state.
State 1 “Powered ascent” is characterized by a constant upward acceleration of 2.25 m/s2. It begins with the rocket launch at t1i = 0 s, y1i = 0 m, v1i = 0 m/s, where the number of the subscript corresponds to the high-level state and the letter (“i” or “f”) corresponds to the initial or final value at the transition that begins/ends the high-level state. State 1 ends at the transition corresponding to engine failure (at a height of y1f = 525 m). State 2 “Unpowered ascent” is characterized by an upward (positive) velocity and gravitational acceleration of g = -9.8 m/s2. It begins with the same values of the low-level variables as state 1 ends (no “time travel” -> time is continuous, no “teleportation” -> y is continuous, conservation of linear momentum -> v is continuous). State 2 ends at the transition corresponding to rest at the apex of the flight (v2f = 0 m/s). State 3 “Descent” is characterized by an downward (negative) velocity and gravitational acceleration of g = -9.8 m/s2. It begins at the end of state 2 and ends when the rocket crashes on the launch pad (x3f = 0 m).
Once the student has decomposed the problem into these three high-level states and determined the values of the low-level variables that describe the transitions, they can program STEPP to produce a simulation of the rocket’s flight, with an animation and real-time graphing providing feedback in multiple representations. STEPP also provides warnings if the student programs unphysical transitions. For example, if the student had forgotten that the gravitational acceleration was negative and had incorrectly inputted g = +9.8 m/s2 for state 2, the rocket would accelerate upwards and never come to rest. In this case, the transition v2f = 0 m/s could not be realized and the simulation would never end. The current version of STEPP provides the error message “You’ll never reach that velocity. The acceleration and change in velocity do not agree.” This message is intended to guide the student to correct their mistake without becoming discouraged. Figure 3 shows a screen capture from the end of the STEPP simulation. States 1, 2, and 3 are shown in blue, orange, and green respectively, with state 1 opened to show the low-level variables describing its initial and final transitions. By opening the other states or reading from directly from the graphs, the student can solve the problem: (a) the rocket reaches a maximum height of 646 m at the end of state 2/beginning of state 3 (orange/green peak in top graph), (b) the rocket comes crashing down to the launch pad t3f – t1f = 16.4 s after engine failure at a final velocity of -113 m/s (right end of the green line segment in middle graph), and (c) STEPP has conveniently provided us with the desired graphs.”
- Reviewer 3 mentioned that: “The authors didn’t use any framework that refers to these dimensions.”
- Thank you for this feedback. The authors of this manuscript are members of a large interdisciplinary research team, and the lead author is a social scientist (me) with extensive training and teaching experience in quantitative methods. Our research plan was to solicit specific feedback from our sample of teachers, and we did not plan to reduce our 13 continuous survey items into three scales. However, since these items correspond to three general areas of inquiry typically examined in this type of usability research, we organized our results by those categories. This was intended to make the results easier for our readers to comprehend and the categories were derived from the theoretical perspective on usability testing that we adopted. While we have enhanced the section on usability research to place our research more clearly within the context of other related usability research (see pages 4-5), we were uncomfortable with the request to add a specific and unintended theoretical framework beyond this because it does not accurately reflect our research plan as we see this as a questionable research practice. This is a philosophy of science issue.
- Reviewer 3 stated: Due to a small sample, non-parametric (not parametric) statistics needs to be applied.
- Thank you for this feedback. One of the reasons we used a within subjects’ design was in anticipation of a small sample size. As we state in the general discussion section, it is typical for usability studies to have small samples and we knew at the start of the study that our sample of teachers would be small.
- This is not necessary as the assumptions of the parametric test used to analyze our data have not been violated. Furthermore, the Wilcoxon Signed Rank Test (the non-parametric equivalent to a within subjects’ t-test) is designed for ordinal data and the data we collected is interval data.
- Part of good science communication is using the more standard data analysis techniques unless there is a clear violation of assumptions. That said, given your instance, we ran the equivalent set of non-parametric analyses (Wilcoxon Signed Rank Test) and our results were unchanged. Given that we did not wish to replace the analyses sections especially after Reviewers 1 and 2 recommended publication of the manuscript as is, we have added this information in a footnote.
- Reviewer 3 stated: Had the authors searched the Internet using keywords “reliability” and “correction for attenuation”, they would easily realize the context of this suggestion. Unless the examined individual measures are of satisfactory reliability (e.g., at least 0.70), the use of individual indicators (e.g., “Likes using STEPP”) is not appropriate. The authors didn't report the reliability of the individual measures.
- Thank you for this feedback. As stated above, we are an interdisciplinary team and the lead author on this paper (me) has extensive training and experience in quantitative methodology. A google search will not assist with the difficulties we are having in understanding some of your concerns as we understand the words but do not understand how you are using them. For instance, it appears to us that you would like us to examine more than one type of reliability. We understand what you mean with respect to the scale reliabilities but not the reliability of the individual measures. On the off chance, this pertains to the correction between time 1 and time 2 measures, we have added this information.
- Reviewer's 3 stated: There is no sense to use percentages for a whole comprising 8 units (in this case 8 participants).
- Thank you for this feedback. In the social sciences, percentages are required for reporting data on ethnicity and other participant demographics as it ensure clearer science communication. So, in response to your feedback to the original version of our manuscript, we added the raw numbers as requested but left in the percentages so that the work was accessible to a larger readership. From our perspective, good science communication means not making your readers work harder to understand your work. Furthermore, this is consistent with the way we report the other demographic data in the participants section.
- Reviewer 3 comment (the most critical issue): This is not done. The authors only used references related to the sample applied. Without this examination in the Discussion part (a must for empirical research!), it is hard to estimate to what extent, if any, this contribution may advance the field.
- Thank you for this feedback. We have added this section to the general discussion (see page 13).